# Gene co-expression network analysis to identify critical modules and candidate genes of drought-resistance in wheat

**Liangjie Lv**[1], **Wenying Zhang**[2], **Lijing Sun**[1], **Aiju Zhao**[1], **Yingjun Zhang**[1], **Limei Wang**[1], **Yuping Liu**[1], **Ziqian Li**[1], **Hui Li**[1]*, **Xiyong Chen**[1]*

**1** Institute of Cereal and Oil Crops, Hebei Academy of Agriculture and Forestry Sciences, Crop Genetics and Breeding Laboratory of Hebei, Shijiazhuang, China, **2** Institute of Dryland Farming Hebei Academy of Agricultural and Forestry Sciences, Hengshui, China

\* chenxiyong369@126.com (XC); zwslihui@163.com (HL)

**Data Availability Statement:** The RNA sequencing data are available from the NCBI Sequence Read Archive under the Bioprojects PRJNA380841 and PRJNA369686. Samples are descried by

## Abstract

### Aim

To establish a gene co-expression network for identifying principal modules and hub genes that are associated with drought resistance mechanisms, analyzing their mechanisms, and exploring candidate genes.

### Methods and findings

42 data sets including PRJNA380841 and PRJNA369686 were used to construct the co-expression network through weighted gene co-expression network analysis (WGCNA). A total of 1,896,897,901 (284.30 Gb) clean reads and 35,021 differentially expressed genes (DEGs) were obtained from 42 samples. Functional enrichment analysis indicated that photosynthesis, DNA replication, glycolysis/gluconeogenesis, starch and sucrose metabolism, arginine and proline metabolism, and cell cycle were significantly influenced by drought stress. Furthermore, the DEGs with similar expression patterns, detected by K-means clustering, were grouped into 29 clusters. Genes involved in the modules, such as dark turquoise, yellow, and brown, were found to be appreciably linked with drought resistance. Twelve central, greatly correlated genes in stage-specific modules were subsequently confirmed and validated at the transcription levels, including *TraesCS7D01G417600.1 (PP2C)*, *TraesCS5B01G565300.1 (ERF)*, *TraesCS4A01G068200.1 (HSP)*, *TraesCS2D01G033200.1 (HSP90)*, *TraesCS6B01G425300.1 (RBD)*, *TraesCS7A01G499200.1 (P450)*, *TraesCS4A01G118400.1 (MYB)*, *TraesCS2B01G415500.1 (STK)*, *TraesCS1A01G129300.1 (MYB)*, *TraesCS2D01G326900.1 (ALDH)*, *TraesCS3D01G227400.1 (WRKY)*, and *TraesCS3B01G144800.1 (GT)*.

### Conclusions

Analyzing the response of wheat to drought stress during different growth stages, we have detected three modules and 12 hub genes that are associated with drought resistance mechanisms, and five of those genes are newly identified for drought resistance. The

Biosamples SAMN06649910-SAMN06649915, SAMN06649934-SAMN06649952 and SAMN06291347-SAMN06291364.

**Funding:** This study was financially supported by the HAAFS Agriculture Science and Technology Innovation Project (2019-4-6-02;4-09-04-01) to LL; Special Foundation of Hebei Academy of Agriculture and Forestry Sciences (2018060303) to LL; Earmarked Fund for Hebei Wheat Innovation Team of Modern Agro-industry Technology Research System (HBCT2018010201) to HL; Key Research and Development Program of china (CN) (2017YF00100603) to LL; Natural Science Foundation of Hebei Province (C2020301004) to LL. We have received funds for covering the costs to publish in open access. The funders had no role in study design, data collection and analysis, decision to publish, or preparation of the manuscript.

**Competing interests:** The authors have declared that no competing interests exist.

references provided by these modules will promote the understanding of the drought-resistance mechanism. In addition, the candidate genes can be used as a basis of transgenic or molecular marker-assisted selection for improving the drought resistance and increasing the yields of wheat.

## Introduction

Wheat is one of the essential commodity crops feeding 20% of the population world [1, 2], and provides critical protection for a healthy global diet [3]. The primary reason why the global yields of wheat have merely been enhanced by 1.0% per year is majority wheat growing in rainfed environments [4]. Drought and heat stress negatively influence the yield and quality of wheat at the critical stages of plant development (flowering and grain formation) [5–8]. With the increasing impact of heat and drought stresses in major wheat-producing belts, the annual wheat supply is likely to be challenged, reducing an estimated 40% per year [5, 9]. Drought and adverse abiotic stress limit wheat productivity and quality in wheat growth and development [10, 11]. With the increasing scarcity of water resources, bread wheat (*Triticum aestivum L.*) has suffered various physiological and biochemical damage in the major wheat-growing region of North China [12, 13]. Some researchers believe that under drought stress, the cultivars with more tillers, more advanced rooting and larger leaf areas had strong drought resistance and could maintain relatively high yield. Drought-resistant wheat varieties can alleviate the changes in stomatal closure, growth inhibition, photosynthesis and hormone composition caused by drought through osmotic regulation, reactive oxygen scavenging system and stomatal closure [14, 15].

The majority of the stress resistance traits in wheat are complicated polygenic and growth stage-specific [16], therefore approaches like phenotyping methods, quantitative trait locus (QTL) mapping, marker-aided selection, allele mining and genetic transformation are employed to develop the drought tolerance of wheat [17]. In the last decade, drought tolerance was researched in genetic bases and involved genes in the genome-wide association study (GWAS) and high-throughput genotyping. Gene microarrays and RNA-sequencing data (RNA-Seq) have also been used as primary tools in the past for gene expression studies of wheat drought stress [18–20]. It is necessary to investigate the molecular mechanisms that respond to drought stress and this should be surveyed at the whole-genome level [21]. Several genes involved in drought stress have been detected at different growth periods, including seedling, tillering, booting, flowering and grain-filling. Although drought stress at the seedling stage also affects the reproductive development and yield of wheat [22], there are fewer study reports in these periods. These genes might serve as valuable genetic resources for wheat drought tolerance improvement.

Transcriptomic and proteomic are also used to analyze wheat drought resistance. A recent transcriptome study showed that over 300 differentially expressed genes (DEGs) of wheat drought were detected in key biological processes, including floral development, photosynthesis, and stomatal movement [23]. Peremarti [24] studied mutant durum wheat under drought stress via transcriptomic and proteomic, and revealed that the regulation of multiple gene expression involving photosystem components, antioxidant enzymes, carbohydrate metabolism enzymes, and tricarboxylic acid cycle enzymes is extraordinarily crucial for wheat drought resistance study. Heat shock protein, cytochrome P450, dehydrated protein, protease inhibitors, and glutathione transferase are commonly involved in wheat drought stress. Transcription factors, such as bZIP, bHLH, ERF, NAC, HD-ZIP, and WRKY, are a category of genes that play an essential role in regulating drought tolerance in wheat [25].

In a semi-arid rainfed agricultural region, the ability of crops to acquire water from depleting soil is critical to the stability of yields. Augmented root growth and depressed shoot growth (increasing the root: shoot ratio) were manifested closely related to the response of wheat development to drought stress [26, 27]. The ability of water extraction from the subsoil and the yield benefits will be improved by faster root embedding and efficient roots increasing [28]. In addition, Rauf [29] observed that the root-shoot ratio of wheat ascends by 50% under drought stress. Abscisic acid (ABA) has been proved to feature as a growth-promoter of roots and simultaneously a growth-inhibitor of shoots [27]. Thus, deep root systems have the potential to improve yield potential in response to drought conditions [30].

With the recent development of bioinformatics, the weighted gene co-expression network analysis (WGCNA) approach has been invented to explore the system-level functionality of the transcriptome [31]. Based on the approach that genes with close functional relationships or distributed in related pathways may have similar expression profiles, WGCNA established a correlation gene network. The genes grouped in a module may have the same function or contribute to similar biological regulation [32]. Then, WGCNA analyzed the interrelation between modules and sample traits and identified the most central and vital hub genes, which supports an efficacious method to explore the in-depth mechanism of complex traits [33].WGCNA has been used extensively to identify gene modules correlated with the identification of putative transcriptional regulation in plants [34–36]. Zheng [37] contrasted transcriptome between wild-type maize and mutants, and used WGCNA constructed from hundreds of transcriptome data to realize the expression pattern of the epidermal wax pathway. The cloned gl14 gene was confirmed responsible for epicuticular wax biogenesis. Through co-expression network analysis, Gamboa-Tuz [38] also identified several abiotic stress-related hub genes in papaya from roots and leaves under drought stress. However, WGCNA has not been adapted to systematically research hub genes correlated with drought stress in wheat. The regulation patterns and molecular mechanisms of drought stress genes have not been comprehensively studied. Therefore, we performed differential expression analysis after WGCNA to identify the gene modules correlated with drought stress in wheat and to forecast the target genes. Since gene expression changes in different environments [39], we executed a transcriptome analysis of different wheat varieties after exposure to drought stress at different growth periods, and investigated the biological function of DEGs at seven different growth stages. In addition, we picked out 12 DEGs for RT-qPCR validation using a drought tolerance variety at different drought times. Our results provide a theoretical basis for further understanding of the molecular response to drought stress in wheat and provide essential information for future genetic improvement and breeding.

## Materials and methods

### Plant material and experimental design

In June 2017, the seeds were planted in Shijiazhuang, Hebei province, China (114.84<E, 37.59<N, 50 meters altitude). The plants were divided into two blocks (A and B), the A block was irrigated twice as a control, while the B block was not irrigated, and other management conditions and techniques were consistent. Based on the investigation of 211 wheat varieties frequently grown in the North China Plain, Jimai 418, drought-resistance wheat cultivated by Hebei academy of agriculture and forestry sciences, was selected as test material. Wheat seedlings cultured in Hoagland nutrient solution for 10 days were subjected to drought (20% PEG-6000) treatment, and leaves of the same part were taken at 0,1,3,6, 12,24 h. The samples were collected from three separate plants and pooled for RNA extraction. We sampled three biological replicates for each material and stored them in liquid nitrogen (-80<C).

## Data collection and preprocessing

The RNA-sequencing data of wheat drought stress were downloaded from the Sequence Read Archive (SRA) (https://www.ncbi.nlm.nih.gov/sra), which were generated from multiple tissues and stages including root, leaves, crown, booting, flowering, grain formation, and anther differentiation. The Illumina HiSeq platform generated the sequencing data. The bioProjects PRJNA380841 and PRJNA369686, which contained 42 datasets mRNA expression data(three technical replicates), were used to estimate the module as an independent cohort. All data in this study were named consistent with the SRA database. Duplicated reads and the bam alignment results of each sample were removed and merged using Picard-tools (v1.41)and samtools (v0.1.18). Raw files were filtered using GATK standard method, and SNPs with distances greater than five were retained. The RPKM of each sample mRNA was calculated using HTseq v0.6.1 [40].

## Differentially expressed gene identification and Gene Ontology (GO) analysis

To identify the genes associated with wheat drought stress, transcriptome sequencing (RNA-seq) was used to compare transcriptome gene expression profiles in different tissues and stages. The R package, edgeR v3.16, was employed to identify the differentially expressed genes between drought stress and normal samples [41]. Genes with a p-value less than 0.05 and absolute value |log2 ratio $\geq 2$| were considered to be differentially expressed genes between different groups [33, 42]. K-means clustering of DEGs was performed using the Euclidean distance measure and complete linkage method. Analyzing each group of DEGs acquired by the transcriptomic, significantly enriched Gene Ontology (GO) terms were detected using Blast2GO (https://www.blast2go.com/). The Kyoto Encyclopedia of Genes and Genomes (KEGG) was enriched with functional annotations using the DAVID database [43].

## Weighted gene co-expression network construction and hub genes identified

The gene co-expression network was constructed by using a WGCNA R package (V 1.51) to further elucidate the functions and mechanisms of genes in wheat drought resistance. The DEGs of 42 samples of different organizations and stages were used to structure the network. The correlation between all gene pairs was calculated to establish a similarity matrix. Using a range of soft threshold values, the average connection network and the fitting evaluation network of scale-free topology models showed an approximate scale-free topology. The adjacency was converted into a topological overlap matrix (TOM), and all coding sequences were hierarchically clustered by TOM similarity. The dynamic tree cut method, which merged highly correlated modules using a height-cut less than 0.25, was used to determine the co-expression gene modules of the gene dendrogram. Finally, we confirmed the stage-specific modules according to the distinguished interrelationship between module membership (MM) and the significance gene (GS). By using Cytoscape v3.4.0, the central and highly connected genes of specific modules in each stage were identified by visualizing the top 150 genes.

## Validation of DEGs by quantitative real-time PCR

The expression of 12 selected genes was verified by quantitative real-time PCR. According to the user manual, the first cDNA strand fragments were synthesized from total RNA using the PrimeScript™ RT Master Mix kit (Takara, Japan). Primer Premier 6.0 was applied to design specific primers, and the housekeeping gene was β-actin. The qRT-PCR was performed on an

ABI7500 real-time fluorescent quantitative PCR instrument (Applied Biosystems, Foster City, USA). The relative expression levels of the candidate genes were measured using the comparative threshold cycle ($2^{-\Delta\Delta CT}$) method.

## Statistical analysis and data visualization

All data analyses and graphics drawing were accomplished in R 3.4.467. Tukey's test of one-way ANOVA was used to compare the physiological parameter differences among different treatments in statistics. Graphics and Heatmaps were plotted with the ggplot and the heatmap packages of R language respectively. Venn diagrams were drawn with the Interactivenn website (http://www.interactivenn.net/). The standardized cluster analysis of transformed gene expression values of log2 (FPKM+0.1) was carried out by the Z-score method. The normalized score of the initial FPKM less than 0.1 was transformed to the lowermost regularised value for all data points. The Pearson-related R software package pheatmap (https://cran.r-project.org/web/packages/heatmap3/index.html) was used to analyze hierarchical clustering and heatmaps.

## Results

### Data collection and preprocessing

The Raw data were reprocessed employing R-packet for background correlation and standardized. The R-packet annotation was used to map the probe to the gene symbol and delete the probe that matched multiple genes. The median value of gene matching numerous probes was used as the eventual expression value. In the end, a total of 1,908,565,550 raw data were obtained from 42 samples. All sequencing data of 42 cDNA libraries fulfilled the quality criteria and satisfied the requirements for subsequent analysis. After assembling and cleaning the data, 1,896,897,901 (284.30 Gb) clean reads were acquired. The clean reads proportion of raw reads ranged from 98.51% to 99.74% in each library, and the ratio of reads with a Phred quality value larger than 30 was between 94.25% and 96.10%. The median GC content of clean reads was 52.61% in 42 samples, and distribution was normal. Then, clean reads were compared with the wheat reference genome (http://plants.ensembl.org/Triticum_aestivum/Info/Index). The relevant Pearson correlation coefficient (PCC) between the biological replications at the same differentiation stage reveals that the sample preparation was quite reproducible.

### Differential expression of genes and functional annotation in wheat drought resistance

A total of 35,021 differentially expressed genes were found between paired contrast (C vs A; CR vs. DR; CL vs DL; CF vs DF; 6C vs. 6A; and 9D vs. 9C) among the samples collected from different wheat organizations and periods (Fig 1A). The number of DEGs among five comparisons (CR vs. DR; CL vs DL; CF vs DF; 6C vs. 6A; and 9D vs. 9C) was performed statistically (Fig 1B), and the number of DEGs decreased gradually as the better differentiation grade of wheat progressed. A total of 35,021 DEGs were recognized in GO and KEGG pathway enrichment analysis using significant enrichment at P-value < 0.05. Within the biological process, the top three enriched GO terms were secondary metabolite process, nucleosome assembly, and peptide biosynthetic process. In terms of molecular function, the most abundant terms were glutathione transferase activity, protein heterodimerization activity, and hydrolase activity. The top three items at the cellular component level were nucleosome, chloroplast part, and plastid part. Pathway analysis exhibited that the differentially expressed genes were mainly enriched in the pathways of cytochrome P450 metabolism, glutathione metabolism, cell cycle, phenylalanine metabolism and carbon fixation in photosynthetic organisms (p < 0.05).

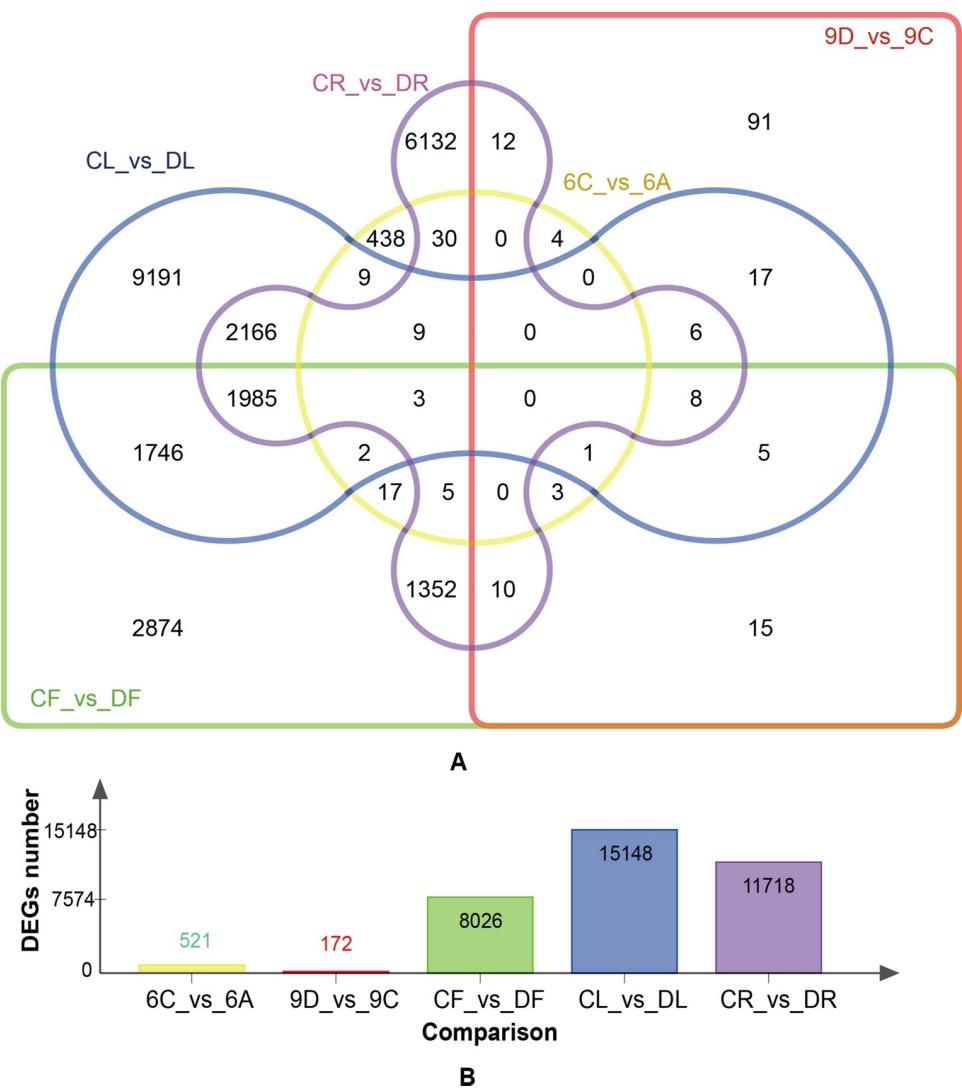

**Fig 1. Venn diagram of differentially expressed genes (DEGs) at different tissues.** (A) Venn diagram of five comparisons of common DEGs (CR vs. DR; CL vs. DL; CF vs. DF; 6C vs. 6A; and 9D vs. 9C). (B) Histogram of five comparisons of the number of DEGs. CR stands for roots under control, DR stands for roots under drought, CL stands for leaves under control, DL stands for leaves under drought, CF stands for flowering stage under control, DF stands for flowering stage under drought, 6C stands for booting stage under control, 6A stands for booting stage under drought, 9C stands for filling stage under control, 9D stands for filling stage under drought.

Furthermore, several pathways related to wheat drought resistance were also enriched, including those of heat shock protein, MYB protein, and serine/threonine-protein kinase, Cytochrome P450, and dehydrin.

## Co-expression network and module construction

Sample correlation analysis was performed on 42 samples showed good reproducibility in the same sample and a significant difference between the samples, which were suitable for WGCNA (Fig 2A). WGCNA can recognize modules of highly related genes, and summarize the modules using module eigengene (ME) or intra-modular hub genes [44]. The choice of soft thresholding value is a critical step to construct WGCNA, so we executed a network topology research of 1 to 20 soft thresholding power, and determined the relative equilibrium of

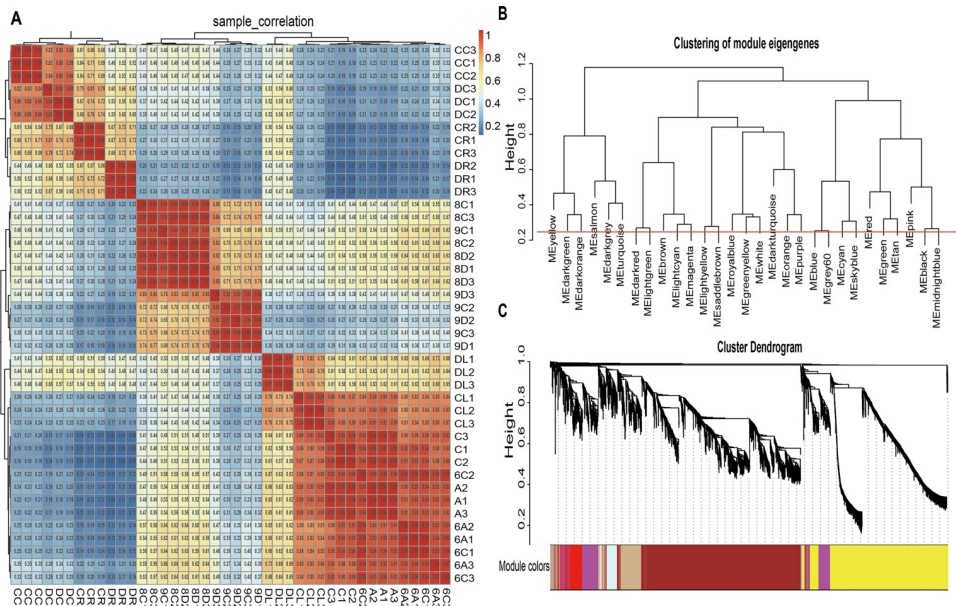

**Fig 2. Correlation between samples and clustering of modules and DEGs.** A: Heat map of the duplicate samples of wheat drought resistance. The colors ranging from blue to red represent Pearson correlation coefficients ranging from 0.2 to 1, indicating low to high correlations, respectively. All samples at the same stage are highly correlated indicating reproducibility of the samples. B: Clustering of module eigengenes; C: Clustering dendrograms of DEGs. Based on the dynamic hybrid branch cutting method, color strips were used for simple visualization of module assignment (branch cutting).

scale independence and the average connectivity of WGCNA. Threshold 9, the minimum value for the scale-free topological fit index on 0.9, was selected as the power value to conduct a hierarchical clustering tree of genes. In the present study, the modules, appointed a unique colored label under the cluster tree, were detected clusters of the co-expressed gene by a co-expression network analysis of 35,021 DEGs. Setting MEDiss Thres to 0.25 to merge parallel modules, 29 modules were identified (Fig 2B). The co-expression module is not independent, but is related to the character and obtained from the whole network. Interaction analysis among the drought feature modules was used to reveal the correlation between the drought feature and the module feature genes. Cluster analysis was carried out in accordance with the characteristic genes in the module to explore the correlation between modules and genes in the module. The results showed that 29 modules were divided into five categories: the first included the yellow, dark green, dark orange, turquoise, dark grey and salmon modules; the second encompassed the dark red, light green, brown, light cyan and magenta modules; the third contained the light yellow, saddle brown, royal blue, green yellow, white, dark turquoise, orange and purple modules; the fourth group comprised the blue, grey60, cyan and sky blue modules; and the fifth group included the red, green, tan, pink, black and midnight blue modules(Fig 2C). Gene modules appointed to the same class may have the same or similar functions and regulatory mechanisms, ranging in size from 74 genes in the saddle brown module to 6530 genes in the turquoise module (Table 1).

## Identification and visualization of stage-specific modules

This paper analyzed the interaction of 29 modules and drew the heatmap (Fig 3A). The independence of each module indicated the absolute independence between the modules and the relative independence of gene expression between the modules. In addition, according to the

**Table 1. List of module sizes.**

| Module | Gene No. | Module | Gene No. | Module | Gene No. |
|---|---|---|---|---|---|
| Black | 1 581 | Green yellow | 893 | Purple | 1 007 |
| Blue | 5 170 | Grey | 695 | Red | 1 914 |
| Brown | 3 369 | Grey60 | 463 | Royal blue | 359 |
| Cyan | 694 | Light cyan | 477 | Saddle brown | 74 |
| Dark green | 314 | Light green | 380 | Salmon | 749 |
| Dark grey | 233 | Light yellow | 448 | Sky blue | 110 |
| Dark orange | 109 | Magenta | 977 | Tan | 868 |
| Dark red | 331 | Midnight blue | 569 | Turquoise | 6 530 |
| Dark turquoise | 277 | Orange | 199 | White | 116 |
| Green | 2 213 | Pink | 1 575 | Yellow | 2 327 |

correlation of eigengenes, the similarity co-expression of all modules was clustered, and 29 modules were principally assigned to five categories. Consistent results were confirmed by the clustering heat maps drawn according to adjacency relations (Fig 3C).

We mapped the heat map of 29 modules and all genes in 42 wheat samples (Fig 3B). From the figure, we could find that the unique module gene was higher expressed in different samples, while other specific module genes were only over-expression in one sample. We focused

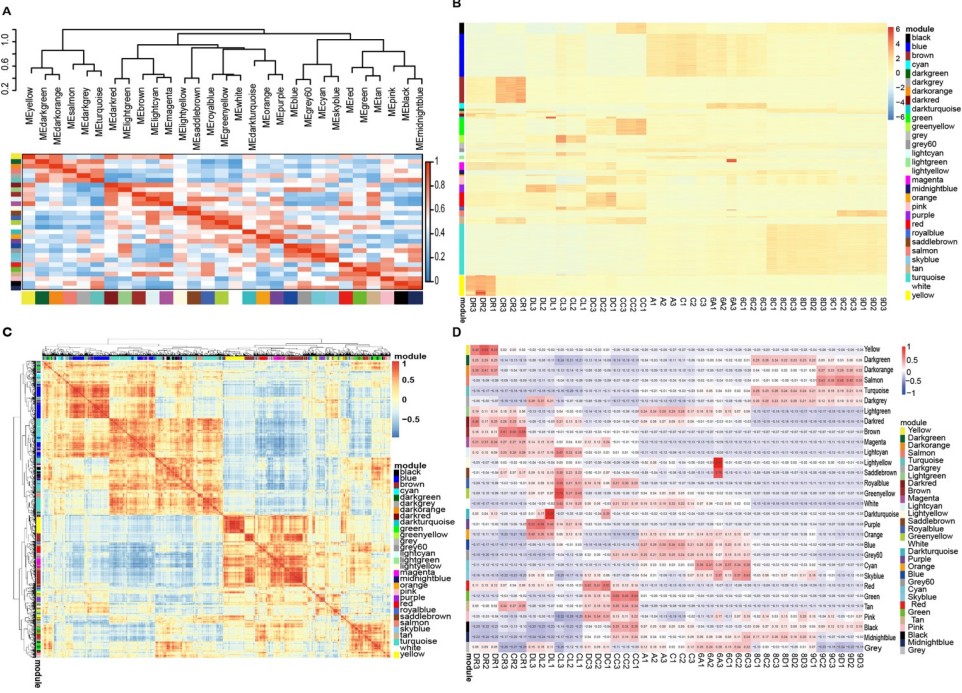

**Fig 3. Visualization of wheat drought resistance co-expression network construction.** (A) Clustering of modules based on eigengenes. The colors ranging from blue to red represent Pearson correlation coefficients ranging from 0 to 1, indicating low to high correlations, respectively. (B) Heat map of correlations between modules and genes. The colors ranging from blue through white to red indicate low through intermediate to high correlations, respectively. (C) 35,021 gene and module correlation clustering diagrams. The colors ranging from blue to red represent Pearson correlation coefficients ranging from—1 to 1, indicating low to high correlations, respectively. (D) Heat map of correlations between modules and different stages. The colors ranging from blue through white to red indicate low through intermediate to high correlations, respectively.

on the analysis of the heatmap between different modules and samples(Fig 3D). We could find that some modules showed significant differences in many materials, such as the high expression of tan modules in CC and CR, manifesting that the tan module was associated with the crown and root development of the control materials. The crown was associated with the root gene; the red module was highly expressed in DC, DL, and DR, indicating that the red module was associated with the crown, leaf, and root ontogeny of the drought-resistant materials. Some modules only showed differences in specific materials, such as the higher expression of the green module in CC, demonstrating that the green module was only related to the crown gene of the control material.

## Analysis of specific drought-resistant related modules

The K-means clustering based on the complete linkage of the Euclidean distance method was used to cluster 35,021 DEGs (Fig 4). Nine stage-specific clusters were determined and plotted according to the expression patterns of genes (Fig 4). Among them were 2,213 genes in the green module, 2,327 genes in the yellow module, 477 genes in the light cyan module, 1914 genes in the red module, 277 genes in the dark turquoise module, 3,369 genes in the brown module, 1,007 Genes in the purple module, 893 genes in the green yellow module, and 109 genes in the dark orange module. The brown, green yellow, green, and light cyan modules were negatively correlated with the root, leave, and crown stages in wheat drought resistance, respectively (Fig 4). In contrast, the yellow, dark orange, purple, and red modules were appreciably positively associated with the root, leaf, and crown stages (Fig 4), respectively. These

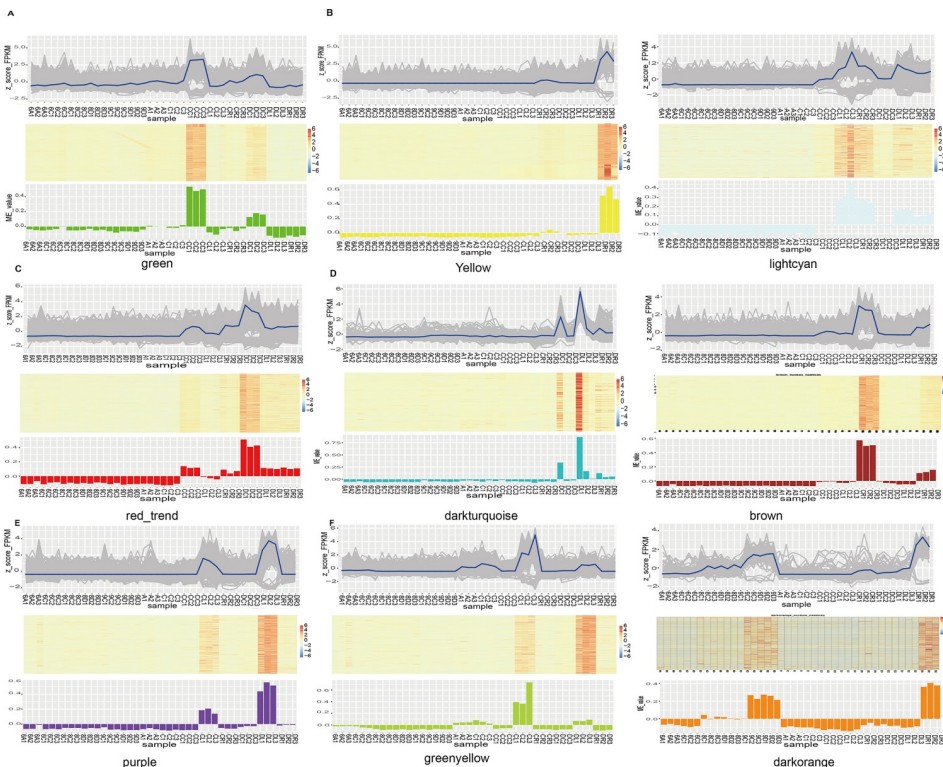

**Fig 4. Visualization of gene expression pattern in k-mean clustering transformation clusters and gene expression levels of significant modules.** Significant differences in gene expression patterns indicate that the modules are related to specific stage development. Clustering heat maps and bar plots represent the level of gene expression in each module. In the heatmap, the colors from blue to red represents low to high expression level, respectively.

results suggest that the gene in the above module was mainly up-regulated and down-regulated in root, leaf and crown. Furthermore, the light cyan module was inversely correlated with the crown and root, which indicates that the genes of this module were significantly down-regulated both in the crown and root period. The genes of these modules might play a central role in wheat drought resistance.

## GO and pathway analysis of genes in specific modules

The functions of genes in specific modules at different periods and the essential signaling of metabolic networks were respectively revealed by GO analysis and pathway analysis in wheat drought resistance. Enriched GO terms of cellular component and biological process were mainly located in the cytoplasm, plastid, cytosol, chloroplast, response to stimulus, response to stress, and peroxisome. These results propose that these cell organizations have a fundamental role in drought stress and that the cytoplasm, peroxisome and chloroplast may be more susceptible to drought stress treatment in wheat during early developmental periods. GO analysis showed that UDP-glycosyltransferase activity, glucosyltransferase activity, oxidoreductase activity, and peroxidase activity were enriched in molecular function. Six KEGG pathways-photosynthesis, DNA replication, glycolysis/gluconeogenesis, arginine and proline metabolism, starch and sucrose metabolism and cell cycle were appreciably influenced by drought stress. The results of enrichment analysis of WGCNA have matched the DEG conclusions, demonstrating that the results were reliable.

## Identification of central and highly connected genes

In this paper, identifying the hub genes in the nine modules, the top 150 connections genes of each particular cluster were analyzed and visualized through Cytoscape 3.4.0 to construct the network diagram, and 38 genes were selected for further analysis by ordering the node degree of candidate genes (Fig 5). The first two genes were investigated as center genes in each module. After filtering, 12 center genes were identified among eight closely related modules of yellow, brown, dark turquoise, green yellow, green, dark orange, light cyan and purple. This suggested that *TraesCS5B01G565300.1*, *TraesCS6B01G425300.1*, *TraesCS7A01G499200.1*, *TraesCS4A01G118400.1*, *TraesCS7D01G417600.1* and *TraesCS2D01G326900.1* might play crucial roles in root at the seedling period. In the dark turquoise module, heat shock protein (HSP) family member 26 (*TraesCS4A01G068200.1*) and HSP90 (*TraesCS2D01G033200.1*) demonstrated the extremely connected mRNA, suggesting their presumed primary missions in the regulation of drought-resistant. The LRR protein 8 (*TraesCS2B01G415500.1*) and MYB protein (*TraesCS1A01G129300.1*) shared the maximum number of nodes in the Green module and were observed to be essential factors in the crown at the seedling stage. *TraesCS3B01G144800.1* is likely an important gene in wheat leaves during drought because it is highly correlated with other proteins in the purple module.

## Validation of DEGs by qPCR

Quantitative real-time PCR (qPCR) was executed to verific the hub genes *TraesCS7D01G417600.1 (PP2C)*, *TraesCS5B01G565300.1 (ERF)*, *TraesCS4A01G068200.1 (HSP)*, *TraesCS2D01G033200.1 (HSP90)*, *TraesCS6B01G425300.1 (RBD)*, *TraesCS7A01G499200.1 (P450)*, *TraesCS4A01G118400.1 (MYB)*, *TraesCS2B01G415500.1 (STK)*, *TraesCS1A01G129300.1 (MYB)*, *TraesCS2D01G326900.1 (ALDH)*, *TraesCS3D01G227400.1 (WRKY)*, *and TraesCS3B01G144800.1 (GT)*. Jimai 418 wheat treated with PEG at different times was used for qPCR validation. These results suggest that, with the increase of Drought stress time, the expression levels of *TraesCS7A01G499200.1* appeared to decrease; the expression of *TraesCS7D01G417600.1*, *TraesCS4A01G068200.1*,

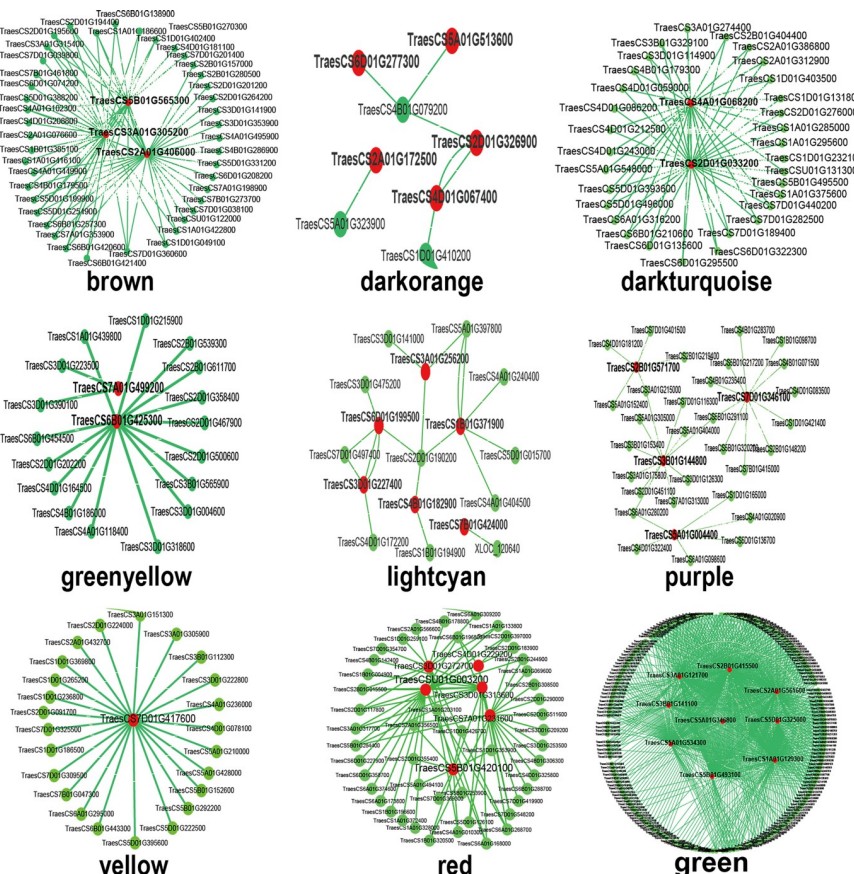

**Fig 5. Visualization of connections of genes in various modules.** Red-colored nodes suggest their central role in the network.

*TraesCS2D01G033200.1*, and *TraesCS2D01G326900.1* tended to increase after drought stress. After drought treatment, the expression of *TraesCS4A01G118400.1* decreased from 3 h to 12 h and then gradually increased; the expression of *TraesCS6B01G425300.1* and *TraesCS3D01G227400.1* decreased progressively, over the first 6 h and then gradually increased (Fig 6). Through RT-qPCR verification and analysis, it was found that *TraesCS7D01G417600.1 (PP2C)*, *TraesCS4A01G068200.1 (HSP)*, *TraesCS2D01G033200.1 (HSP90)*, *TraesCS7A01G499200.1 (P450)*, *TraesCS2D01G326900.1 (ALDH)* were identified genes closely related to wheat drought resistance, and speculated that genes in the root related yellow module and the leaf related dark turquoise module were involved in the wheat drought resistance response.

## Discussion

### The response of plants to drought stress in different growth periods

Drought between the seedling stage and the filling stage continuously affects wheat growth, development, quality, and yield. Therefore, we performed drought treatment on wheat materials in different stages and tissues to analyze DEGs. Previous studies have found many DEGs that affect the early reproductive stages of wheat under drought stress, but the functions of these genes are mostly unknown. A study by Ugarte et al. [45] showed that high temperature and drought between the booting and anthesis stages reduced wheat yield by 46% compared to between the heading and anthesis stages. Therefore, the effect of drought stress on wheat yield

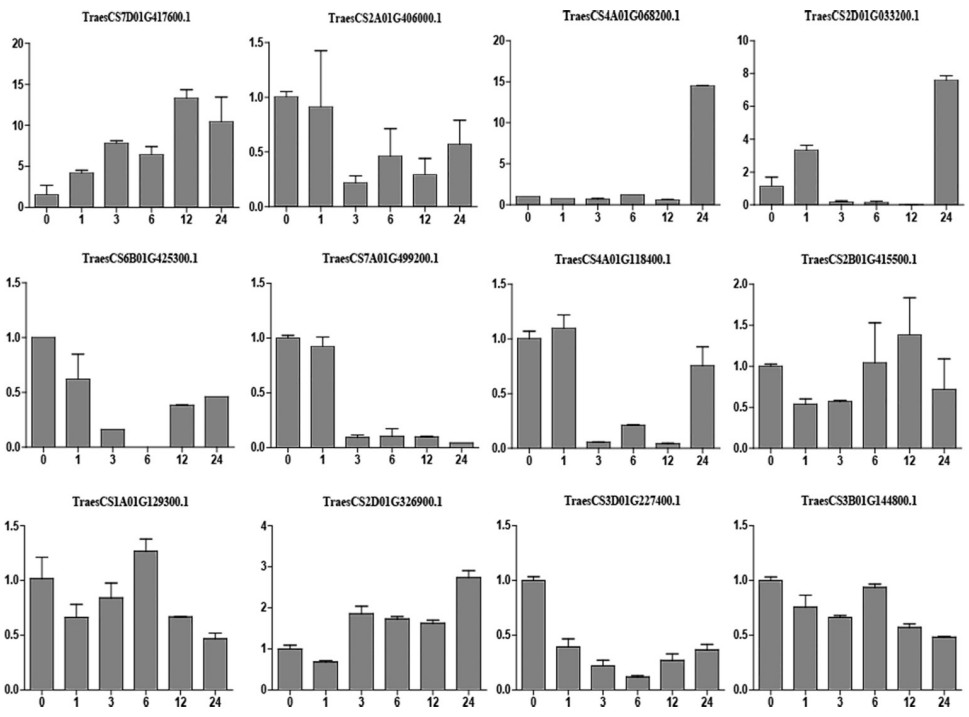

**Fig 6. Validation of hub genes using RT-qPCR.**

during the early reproductive period may be greater than that experienced during anthesis. We detected 521 DEGs during the booting stage. Drought at the booting stage resulted in the death of wheat florets and spikelets, and drought during the anthesis and filling stages reduced wheat grain size and weight [46]. We found 8026 DEGs during wheat anthesis under drought stress. The principal genes were distributed in the green (2213) and red (1914) modules. We analyzed 13 genes that were most highly related to the top 200 genes, of which four were validated. Most of the genes functioned in anther formation, regulation of flowering period, and pollen viability. Drought stress during grain filling and anthesis can gravely diminish wheat production, which primarily manifests as a decreased number of grains and ears, and decreased filling speed [47]. Drought stress had a smaller effect on wheat grain gene expression during the filling stage, and therefore fewer DEGs (172) were detected during the filling stage. The main genes were distributed in the dark orange module. We analyzed five genes that were highly associated with the top 200 genes and validated one of them, which functions as betaine aldehyde dehydrogenase. Plentiful DEGs were identified in the roots and leaves, with 15,148 DEGs in the leaves and 11,718 in the roots. DEGs in the roots were mainly distributed in the yellow, brown, and dark orange modules; DEGs in the leaves were mainly distributed in the light cyan, dark turquoise, purple, and green-yellow modules. The functions of most of these DEGs in wheat are unknown. The changes in these gene expression levels under drought stress may directly influence grain development and yield in wheat. We prepare to research the function and regulatory mechanisms of these genes in future experiments.

## Analysis of candidate genes

Twelve hub genes were identified and validated in this study. A number of these genes have previously been reported by other researchers, confirming the reliability of this validation method, whereas other genes have not been reported, and thus were first discovered in the

present study; their functions will be further elucidated in future work. *TraesCS7D01G417600.1* is a gene in the PP2C protein phosphatase family. Studies have reported that PP2C family genes are closely associated with plant drought resistance [48], and this gene is a new member of the F subfamily of the PP2C family. However, its function is unclear, and we will conduct follow-up experiments to further investigate its involvement in mechanisms of drought resistance. ERF is an important transcription factor in plant drought stress response pathways, similar to *OsERF40*, *OsERF109*, *OsERF4a*, *GmERF123*, and other genes cloned in rice [49–51], and *TaERF1* in wheat [52]. *TraesCS5B01G565300.1* is an ethylene-responsive transcription factor 1B gene, which differs in sequence from the gene reported by Xu et al. [52]. However, a study of drought resistance and dehydration in soybean by Ferreira et al. [53] considered the *ERF1b* gene to be a hormone response factor associated with drought resistance. Heat shock proteins produced by plants under high temperatures can protect proteins in the plant body from damage, or repair damaged proteins, thereby protecting the plant, indicating that the induced production of heat shock proteins can confer plants with drought resistance. There have been many studies on heat shock proteins in wheat, including *TaHSFA6e* [54], *HSP90*, and *HSP70* [55]. *TraesCS2D01G033200.1* is heat shock protein *HSP90*, which is consistent with previous reports. *TraesCS4A01G068200.1* is heat shock protein *HSP26*, and studies have found that the wheat *HSP26* gene helps improve heat resistance [56]. Stressors such as high temperature and drought often lead to changes in membrane lipid fluidity, loss of biological membrane integrity, inactivation of enzyme activity, and osmotic and oxidative stress, so increased heat resistance in wheat corresponds to increased drought resistance. Abscisic acid (ABA) is an essential plant hormone and plays a vital role in plant response to drought stress in wheat. The accumulation of ABA in plants depends on the biosynthesis and metabolism of ABA, so ABA catabolism is also a major process that regulates ABA content in cells. ABA biosynthesis is regulated by protein kinases. *TraesCS2B01G415500.1* is a serine/threonine kinase that plays a vital role in the ABA pathway. ABA is primarily catabolized by glucose conjugation and hydroxylation. The conjugation process is completed by UDP-glucosyltransferase, and the initial hydroxylation of ABA is catalyzed by four cytochrome P450 monooxygenases, CYP707A1-CYP707A4. To date, several studies have identified the transcriptional regulators of ABA metabolism genes, such as the bZIP transcription factor VIP1 [57], but the regulation of ABA catabolism and its effect on plant drought resistance requires further study. TraesCS3B01G144800.1 is a UDP-glucosyl-transferase (UGT), which may regulate the degradation of ABA to enhance the drought resistance of wheat. In a study of the *Arabidopsis thaliana UGT* gene, Chen et al. [58] found that plants overexpressing UGT75B1 downregulated ABA response genes under drought stress, indicating that *Arabidopsis* UGT75B1 responds to abiotic stress through glycosylation of ABA. This gene has not yet been identified in wheat, and we are currently validating its function(s). *TraesCS7A01G499200.1* is cytochrome P450, which also participates in the decomposition of ABA and response to wheat drought stress; this result is consonant with a study by Wei et al. [59]. The MYB protein family is a type of transcription factor that contains the MYB conserved domain, and is widely involved in regulating various physiological responses such as plant growth and stress tolerance. There have been many reports regarding MYB transcription factors regulating drought stress pathways in wheat [60, 61]. *TraesCS1A01G129300.1* is the R1R2R3-MYB protein gene, which is accordant with the results of Dai et al. [62]. *TraesCS4A01G118400.1* is the myb78 gene, which is agreeable with the conclusion of Monika et al. [63]. Members of the WRKY gene family play an extremely critical role in regulating plant responses to drought stress and the establishment of signal transduction pathways. The wheat genes *TaWRKY1*, *TaWRKY2*, *TaWRKY10*, *TaWRKY33*, *TaWRKY40*, *TaWRKY44*, and *TaWRKY93* have been validated to be associated with drought resistance [64, 65]. In the present study, the *TraesCS3D01G227400.1* gene is *WRKY 22*, which has not been proclaimed to link

with drought resistance. *TraesCS2D01G326900.1* is betaine aldehyde dehydrogenase. Wang et al. [66] found that the content of proline and betaine in transgenic wheat leaves is higher, and betaine aldehyde dehydrogenase (BADH) activity is elevated under drought stress. The accumulation of proline and trimethylglycine in plants may play a key role in tolerance to drought stress [67]. *TraesCS2A01G406000.1* is a lysine/histidine transporter, and its function during wheat drought stress is unclear and requires further study.

## Supporting information

**S1 Table. Primers of genes used in RT-qPCR.**
(DOCX)

## Author Contributions

**Conceptualization:** Liangjie Lv.

**Data curation:** Liangjie Lv, Aiju Zhao, Limei Wang.

**Formal analysis:** Liangjie Lv.

**Funding acquisition:** Liangjie Lv.

**Investigation:** Liangjie Lv, Wenying Zhang, Yuping Liu, Ziqian Li.

**Methodology:** Liangjie Lv.

**Project administration:** Liangjie Lv.

**Supervision:** Hui Li, Xiyong Chen.

**Validation:** Liangjie Lv, Lijing Sun, Yingjun Zhang.

**Writing – original draft:** Liangjie Lv.

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
