## [Decision Letter · Decision Letter 0]

27 May 2020

PONE-D-20-08497

Gene co-expression network analysis to identify critical modules and candidate genes of drought-resistant in wheat

PLOS ONE

Dear Dr. chen,

Thank you for submitting your manuscript to PLOS ONE. After careful consideration, we feel that it has merit but does not fully meet PLOS ONE’s publication criteria as it currently stands. Therefore, we invite you to submit a revised version of the manuscript that addresses the points raised during the review process.

Please seek professional proofreading to the manuscript;Address all concerns raised by the two reviewer one by one;

We look forward to receiving your revised manuscript.

Kind regards,

Wujun Ma

Academic Editor

PLOS ONE

Journal Requirements:

2. In your Methods section, please provide additional details regarding the plant materials used in your study and ensure you have described the source. For more information regarding PLOS' policy on materials sharing and reporting, see https://journals.plos.org/plosone/s/materials-and-software-sharing#loc-sharing-materials.

4. We note that you are reporting an analysis of a microarray, next-generation sequencing, or deep sequencing data set. PLOS requires that authors comply with field-specific standards for preparation, recording, and deposition of data in repositories appropriate to their field. Please upload these data to a stable, public repository (such as ArrayExpress, Gene Expression Omnibus (GEO), DNA Data Bank of Japan (DDBJ), NCBI GenBank, NCBI Sequence Read Archive, or EMBL Nucleotide Sequence Database (ENA)). In your revised cover letter, please provide the relevant accession numbers that may be used to access these data. For a full list of recommended repositories, see http://journals.plos.org/plosone/s/data-availability#loc-omics or http://journals.plos.org/plosone/s/data-availability#loc-sequencing.

6. We note you have included a table to which you do not refer in the text of your manuscript. Please ensure that you refer to Table 1 and 2 in your text; if accepted, production will need this reference to link the reader to the Table.

Reviewers' comments:

Reviewer's Responses to Questions

**Comments to the Author**

1. Is the manuscript technically sound, and do the data support the conclusions?

Reviewer #1: Yes

Reviewer #2: Yes

2. Has the statistical analysis been performed appropriately and rigorously? 

Reviewer #1: Yes

Reviewer #2: Yes

3. Have the authors made all data underlying the findings in their manuscript fully available?

Reviewer #1: No

Reviewer #2: Yes

4. Is the manuscript presented in an intelligible fashion and written in standard English?

Reviewer #1: No

Reviewer #2: No

5. Review Comments to the Author

Reviewer #1: Drought stress is an important limiting factor in wheat production. Lv et al analyzed differiential expression genes of wheat between drought stress and normal growth conditions by means of RNA-seq and gene co-expression network analyses. They detected three modules and 26 twelve hub genes that are associated with drought resistance mechanisms, and newly identified five candidant genes for drought resistance. The study is interesting.

However, several main issues need be addressed, and the manuscript need to be carefully revised due to many underexpressing stentences and many wrongtypos/grammars.

For examples,

1 DESCRIBE the treated samples at which growth stages, which tissues in Material, and RESULTS parts and Figure1. Figure1 should explain themeanings of C vs A, CR VS DR, ...

2 Title ,drought-resistant should be corrected to drought-resistance

3. In Abstract and result parts, in the "genes involved in the modules, such as dark turquoise, yellow and brown, were found", dark turquoise, yellow and brown should be explained by expert names, such as detailed genes/pathways.

4. In Abstract, Twelve central, greatly correlated genes in stage-specific modules were

subsequently confirmed and validated at the transcription levels, including TraesCS7D01G417600.1,

TraesCS5B01G565300.1, TraesCS4A01G068200.1, TraesCS2D01G033200.1, TraesCS6B01G425300.1,

TraesCS7A01G499200.1, TraesCS4A01G118400.1, TraesCS2B01G415500.1, TraesCS1A01G129300.1,

TraesCS2D01G326900.1, TraesCS3D01G227400.1 and TraesCS3B01G144800.1, the genes should be written with the gene name, such as PP2C, ERF1B...

5. Lines 26-27, and results, what are five of the genes newly identified for drought resistance in the study? How do you verify their functions in drought resistance?

6. The MS English writing must be carefully improved and throughtly edited by a native English speaking expert.

7. Each figure, SHOULD BE NUMBERED.

Reviewer #2: Major issues

1. The English language needs professional proofreading as there are massive typos and grammar errors in the main document.

2.A short introduction referring to the current understanding of drought-resistant in wheat is suggested.

3. A total of twelve genes (TraesCS7D01G417600.1, TraesCS5B01G565300.1, TraesCS4A01G068200.1, TraesCS2D01G033200.1, TraesCS6B01G425300.1, TraesCS7A01G499200.1, TraesCS4A01G118400.1, TraesCS2B01G415500.1, TraesCS1A01G129300.1, TraesCS2D01G326900.1, TraesCS3D01G227400.1 and TraesCS3B01G144800.1) were validated at the transcription levels, why?

Minor issues

1. Please remove the background in the text. For example, Line 61-70; Line79-101, et al.

2. What is the “follow-up test material”(Line105)?

3. line 110, punctuation missing in the sentence.

4. line 108-109, “During the growth of wheat (the roots, leaves, and crowns), the material was collected from 6 separate plants tissue from each material and pooled for RNA extraction” could be change into “The roots, leaves, and crowns were collected from 6 separate plants from each material and pooled for RNA extraction”

5.Line 150, 2-ΔΔCT should be 2-ΔΔCT.

6. PLOS authors have the option to publish the peer review history of their article (what does this mean?). If published, this will include your full peer review and any attached files.

Reviewer #1: No

Reviewer #2: Yes: Rugen Xu

---

## [Author Response · Author response to Decision Letter 0]

24 Jun 2020

Referee #1:

We want to begin by thanking Referee #1 for writing that “Drought stress is an important limiting factor in wheat production. The study is interesting.” We also appreciated the constructive criticism and suggestion. We addressed all the points raised by the reviewer, as summarized below.

1. Describe the treated samples at which growth stages, which tissues in Material, and Figure1 should explain the meanings of C vs A, CR VS DR, ... (in results).

As the reviewer suggested, in the material method section, we added a description of the growth stage and tissue of the treated samples and added the meaning interpretation of C vs A, CR vs DR… in the legend of Figure 1 on the result section. 

2. Thanks to the referee’s comment, we have changed the “drought-resistant“ to “drought-resistance“ in the title.

3. In Abstract and result parts, in the "genes involved in the modules, such as dark turquoise, yellow and brown, were found", dark turquoise, yellow and brown should be explained by expert names, such as detailed genes/pathways.

Thank you very much for your comments. In the abstract and results section, the module's names were described in dark turquoise, yellow and brown ... This is because when WGCNA is used for gene clustering analysis, genes that are functionally related or similar are grouped and represented by the same color. The same color module contains many genes, which cannot be enriched into a single pathway or explained by expert names. So only colors can be used to describe modules.

4. In Abstract, Twelve central, greatly correlated genes in stage-specific modules were subsequently confirmed and validated at the transcription levels, including TraesCS7D01G417600.1, TraesCS5B01G565300.1, TraesCS4A01G068200.1, TraesCS2D01G033200.1, TraesCS6B01G425300.1, TraesCS7A01G499200.1, TraesCS4A01G118400.1, TraesCS2B01G415500.1, TraesCS1A01G129300.1, TraesCS2D01G326900.1, TraesCS3D01G227400.1 and TraesCS3B01G144800.1, the genes should be written with the gene name, such as PP2C, ERF1B...

As the reviewer suggested, We looked up the names of these genes and added them in parentheses. 

5. Lines 26-27, and results, what are five of the genes newly identified for drought resistance in the study? How do you verify their functions in drought resistance?

Referee #2 asks what are five of the genes and how to verify….. Lines 351-353, we added the specific names of these five genes to the paper. We treated wheat Jimai 418 in the seedling stage with drought (20%PEG) for a different time and extracted RNA for RT-PCR verification. It was found that the expression levels of these five genes were significantly different under drought stress, which was speculated to be related to response to drought stress. We would like to note that we investigated the five genes in the previous study and found no reported that the five specific genes were associated with drought resistance in wheat. Therefore, in this manuscript, we focused on these five genes and will carry out transgenic verification in subsequent experiments.

6. The MS English writing must be carefully improved and thoroughly edited by a native English speaking expert.

As suggested by the reviewer we have invited professional scientific editing service (peerwith) and native expert (Ally Oakes) to polish the language of the article. Please see the attachment for the editing certificate. Our colleagues studying abroad have also made careful improvements to the English expressions throughout the manuscript. All changes have been revised in the manuscript with track changes.

7. Because of the referee’s comment, we have numbered the pictures in the manuscript.

Referee #2:

We would like to thank the referees for their thoughtful review of our manuscript. We believe that the additional changes we have made in response to the reviewer's comments have made this a significantly stronger manuscript. Below is our point-by-point response to the referee’s comments.

Major issues

1. The English language needs professional proofreading as there are massive typos and grammar errors in the main document.

The language of the article has been polished by professional scientific editing service (peerwith) and native expert (Ally Oakes). Please see the attachment for the editing certificate. Our colleagues studying abroad have also made careful improvements to the English expressions throughout the manuscript. All changes have been revised in the manuscript with track changes.

2. A short introduction referring to the current understanding of drought-resistant in wheat is suggested.

According to the referee’s suggestion, we added a brief introduction to the current understanding of drought-resistant in wheat at the end of the first paragraph.

3. A total of twelve genes (TraesCS7D01G417600.1, TraesCS5B01G565300.1, TraesCS4A01G068200.1, TraesCS2D01G033200.1, TraesCS6B01G425300.1, TraesCS7A01G499200.1, TraesCS4A01G118400.1, TraesCS2B01G415500.1, TraesCS1A01G129300.1, TraesCS2D01G326900.1, TraesCS3D01G227400.1 and TraesCS3B01G144800.1) were validated at the transcription levels, why?

According to the k-mean diagram of 9 modules determined in this paper, the drought-treated tissues and developmental stages were compared with the transcriptional results, and eight modules with consistent results were obtained for analysis. The 38 genes with the highest correlation degree were selected by using the protein interaction diagram. After consulting the gene function and referring to previous literature, we speculated that these 12 genes might be related to drought.

Minor issues

1. Thanks to the referee’s comment, the wrong background have removed in the text. 

2. Line105, the “follow-up test material” was changed to “test material”.

3. Line 110, to better understand we have punctuated in this sentence.

4. lines 108-109, we agree with the reviewer's comment and changed to “The samples were collected from 3 separate plants and pooled for RNA extraction.”

5. Line 150, according to the referee’s suggestion, 2-ΔΔCT was changed to 2-ΔΔCT.

---

## [Editor Report · Decision Letter 1]

1 Jul 2020

Gene co-expression network analysis to identify critical modules and candidate genes of drought-resistant in wheat

PONE-D-20-08497R1

Dear Dr. chen,

We’re pleased to inform you that your manuscript has been judged scientifically suitable for publication and will be formally accepted for publication once it meets all outstanding technical requirements.

Kind regards,

Wujun Ma

Academic Editor

PLOS ONE